# Cryo-EM and femtosecond spectroscopic studies provide mechanistic insight into the energy transfer in CpcL-phycobilisomes

Lvqin Zheng[1,2,5], Zhengdong Zhang[1,3,5], Hongrui Wang[1,3,5], Zhenggao Zheng[1,3,5], Jiayu Wang [4], Heyuan Liu[4], Hailong Chen [4], Chunxia Dong[1,3], Guopeng Wang[1], Yuxiang Weng [4] ✉, Ning Gao [1,2] ✉ & Jindong Zhao [1,3] ✉

Phycobilisomes (PBS) are the major light harvesting complexes of photosynthesis in the cyanobacteria and red algae. CpcL-PBS is a type of small PBS in cyanobacteria that transfers energy directly to photosystem I without the core structure. Here we report the cryo-EM structure of the CpcL-PBS from the cyanobacterium *Synechocystis* sp. PCC 6803 at 2.6-Å resolution. The structure shows the CpcD domain of ferredoxin: NADP$^+$ oxidoreductase is located at the distal end of CpcL-PBS, responsible for its attachment to PBS. With the evidence of ultrafast transient absorption and fluorescence spectroscopy, the roles of individual bilins in energy transfer are revealed. The bilin $^{II}\beta^{82}_2$ located near photosystem I has an enhanced planarity and is the red-bilin responsible for the direct energy transfer to photosystem I.

The cyanobacteria are one of the earliest groups of organisms to convert solar energy to chemical energy[1,2]. They are the first organisms that created linear electron transfer with two photosystems, photosystem II (PSII) and photosystem I (PSI), and were responsible for the rise of oxygen content on earth more than 2.4 billion years ago[1–3]. The major light harvesting antennae for capturing solar energy in cyanobacteria and red algae are phycobilisomes (PBS)[4–7], which transfer the absorbed light energy to either PSII or PSI to drive electron transfer. PBS consists of phycobiliproteins (PBP), which have covalently attached pigments called bilins, and linker proteins[4,8]. Two types of PBS exist in cyanobacteria, the CpcG-PBS and the CpcL-PBS. CpcG-PBS consist of two parts: a core and peripheral rods, which are associated with the core through linker proteins CpcG. Recently determined cryo-EM PBS structures from red algae[9,10] and the cyanobacteria[11–13] revealed how linker proteins and PBP are organized into a highly ordered light harvesting architectures. The CpcL-PBS has a much smaller size and consists of only one rod without allophycocyanin core[14–16]. CpcL-PBS, which is attached to thylakoid membranes through the linker protein CpcL (also called CpcG2 in *Synechocystis*

6803 and CpcG3 in *Anabaena* 7120[17]), is associated with PSI[18] and able to transfer absorbed light energy to PSI[16,19]. CpcL-PBS is also involved in the formation of NAD(P)H-dehydrogenase (NDH)-CpcL-PBS-PSI (NDH-PBS-PSI) supercomplex for cyclic electron transfer (CEF)[20] and a recent study shows that the attachment of ferredoxin-NADP$^+$ oxidoreductase (FNR) to the CpcL-PBS is required for CEF[21]. Most cyanobacterial FNR contains three domains: CpcD domain, FAD binding domain, and NADPH binding domain, and the three-domain FNR (FNR$_{3D}$) is attached to the peripheral rods of the cyanobacterial CpcG-PBS and CpcL-PBS[22–24]. However, the attachment of FNR$_{3D}$ to the PBS rods has not been observed in the recently determined cryo-EM structures CpcG-PBS[11–13]. To understand the mechanism for direct energy transfer from CpcL-PBS to PSI and its role in NDH-PBS-PSI supercomplex formation, we determine the cryo-EM structures of the CpcL-PBS with an attached FNR from *Synechocystis* 6803. Combined with picosecond spectroscopic evidence, our study reveals the nature of the terminal emitter-like red-shifted bilin and shed light on how CpcL-PBS could effectively transfer light energy within PBS and to PSI in the absence of a PBS core.

[1]School of Life Sciences, Peking University, Beijing 100871, China. [2]State Key Laboratory of Membrane Biology, Peking University, Beijing 100871, China. [3]State Key Laboratory of Protein and Plant Gene Research, Peking University, Beijing 100871, China. [4]Laboratory of Soft Matter Physics, Institute of Physics, Chinese Academy of Sciences, Beijing 100190, China. [5]These authors contributed equally: Lvqin Zheng, Zhengdong Zhang, Hongrui Wang, Zhenggao Zheng. ✉e-mail: yxweng@iphy.ac.cn; gaon@pku.edu.cn; jzhao@pku.edu.cn

## Results

### Overall structure of CpcL-PBS complex

CpcL-PBS were prepared from a strain of *Synechocystis* 6803 lacking the genes of *apcAB*[24] and the purified CpcL-PBS were analyzed spectroscopically and biochemically for its composition and functional integrity (Supplementary Fig. 1). SDS-PAGE analysis followed by protein N-terminal sequencing revealed that purified CpcL-PBS contained phycocyanin (PC) and linker proteins, including CpcA, CpcB, CpcC1, CpcC2, and CpcL. The CpcL-PBS preparation also contains the three-domain FNR with a molecular mass of 45 kDa (Supplementary Fig. 1e). The absorption spectrum of the isolated CpcL-PBS has a maximum absorption at 620 nm (Supplementary Fig. 1g), typical of a phycocyanin absorption. Its fluorescence emission spectrum at room temperature has two peaks, one at 648 nm and the other at 668 nm, same as reported previously[24,25]. The emission peak at 648 nm is similar to the phycocyanin hexamer with a CpcG linker[26,27], while the emission peak at 668 nm from phycocyanin (PC) has not been observed in other PC trimer or rods (Supplementary Fig. 1g). At 77 K, one major emission peak at 670 nm is observed (Supplementary Fig. 1h). Negative-staining electron microscopy (EM) analysis of the CpcL-PBS sample shows that it contains rod-like structures and the majority of the rods contain three hexamers. Some particles with more than three αβ hexamers were also found (Supplementary Fig. 1f), indicating that CpcL-PBS complexes are heterogenous in nature, which is consistent with a previous work[24]. To minimize the PBS complex disassembly during cryo-grid preparation, samples were treated with 0.012% glutaraldehyde and the particles of CpcL-PBS were automatically selected and classified (Supplementary Fig. 2). A 2.6-Å density map of CpcL-PBS

containing three layers hexamers was obtained, which enabled us to build atomic models for all chromophores (bilins) and protein chains (Supplementary Fig. 3a–d). Notably, the three-layer architecture was the dominant species as 2D and 3D classifications failed to enrich other populations. This three-layered CpcL-PBS complex is 160 Å in length and 100 Å in diameter (Fig. 1a–c). The complex contains 40 subunits, including 18 PC αβ monomers (CpcA-CpcB heterodimer) and three linker proteins, CpcL, CpcC1, and CpcC2 in a 1:1:1 stoichiometry (Fig. 1d, e). The CpcD domain of the FNR$_{3D}$ could be clearly identified at the distal end of the CpcL-PBS (Fig. 1d, e). The four linkers are organized in an order of CpcL-CpcC1-CpcC2-CpcD (of FNR$_{3D}$) in a nearly linear manner with CpcL close to the thylakoid membrane and FNR$_{3D}$ at the distal end (Fig. 1d, e). CpcL-PBS closely resembles the rod of CpcG-PBS from *Synechocystis* 6803[13] with an RMSD of 0.4 Å between them (Fig. 1f). These linkers form a linker skeleton that provides a support for CpcL-PBS assembly and its association with PSI (Fig. 1g).

### The linker binding mode defines the CpcL-PBS assembly

The Pfam00427 domain of CpcL was located in the cavity of the bottom hexamer of CpcL-PBS, the membrane-proximal hexamer. The transmembrane region of CpcL, which was predicted to anchor CpcL-PBS to the thylakoid membranes[15], was not resolved probably due to its unrestricted movement in the purified complexes. Like CpcC proteins in red algae[9,10] and other cyanobacteria[11–13], both CpcC1 and CpcC2 of *Synechocystis* 6803 have two structural domains, Pfam00427 at the N-terminus and Pfam01383 at the C-terminus. The Pfam01383 domain of CpcC1 interacts with Pfam00427 domains of CpcL while

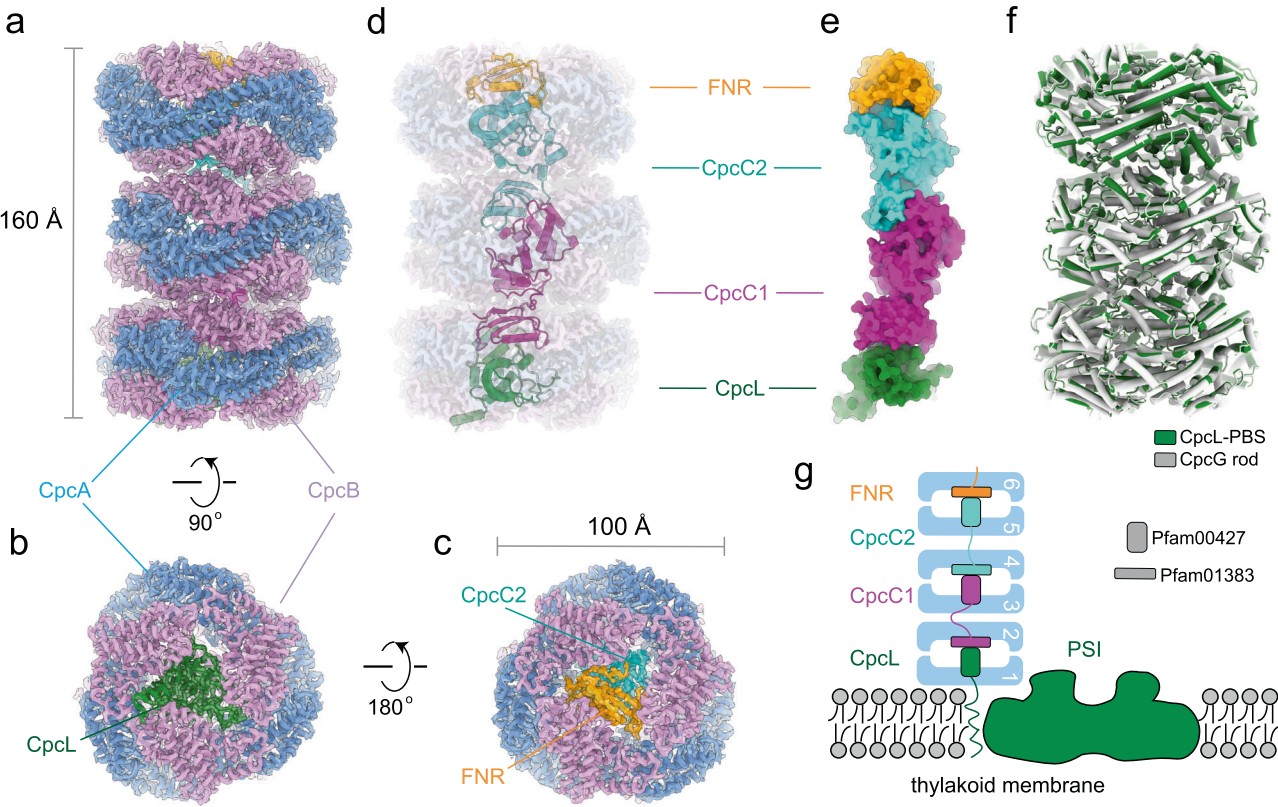

**Fig. 1 | Structure of CpcL-PBS from *Synechocystis* 6803. a–c** Side-view (**a**), bottom-view (**b**), and top-view (**c**) of the CpcL-PBS, respectively. α-PC, β-PC, CpcL, CpcC2, and the CpcD domain of FNR are painted in plum, cornflower blue, forest green, cyan and orange, respectively. **d** The arrangement of linker proteins (in ribbon presentation) within CpcL-PBS. From bottom to top: CpcL, CpcC1, CpcC2, and the CpcD domain of FNR. CpcC1 is in magenta and the other color scheme is same as in (**a–c**). **e** Same as (**d**) except that the linker proteins are in fill-in

presentation and there is no PC background. **f** Structural comparison of CpcG rod (PDB 7SC8)[13] and CpcL-PBS. **g** Schematic presentation of CpcL-PBS and PSI in the thylakoid membrane with the PC hexamers in blue and PSI complex in forest green. The two domains that occupy the central cavities of the hexamers: Pfam00427 (vertical rectangles) and Pfam01383 (horizontal rectangles). PC trimers are numbered from bottom to top sequentially.

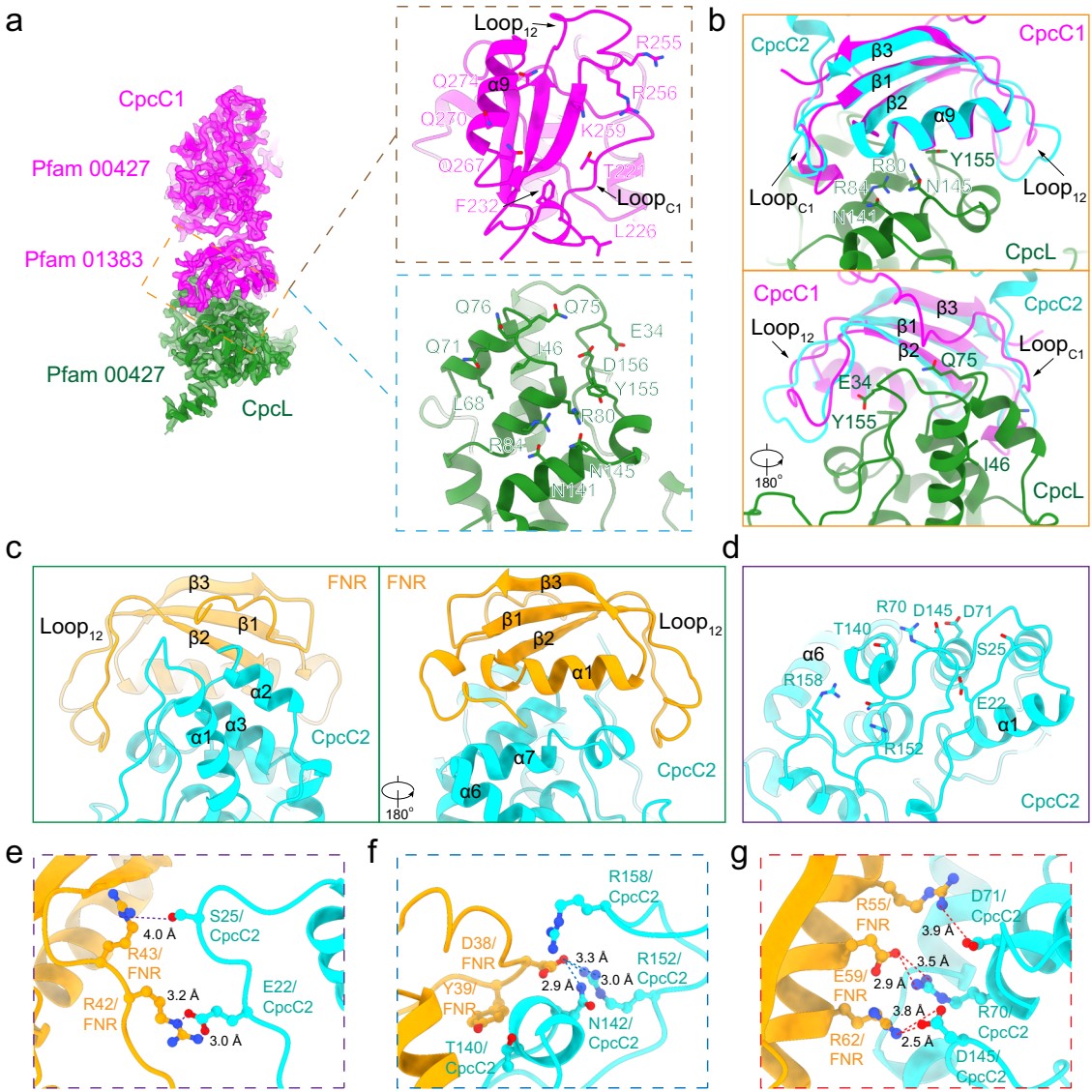

**Fig. 2 | Interactions between the linker proteins in the formation of linker skeleton of CpcL-PBS. a** Interaction of CpcC1 (magenta) with CpcL (forest green). The bottom view of CpcC1 (top panel) and the top view of CpcL (bottom panel) show the spatial distribution of the residues that are involved in the hydrophobic interactions and hydrogen-bonding. $Loop_{C1}$: the loop close to β1 of CpcC1. $Loop_{12}$: the loop between β1 and β2 of CpcC1. **b** Detailed interactions between CpcC1 (magenta) and CpcL (forest green) with the same region of CpcC2 (cyan) superimposed on CpcC1. The lower panel shows the same structures shown in the upper panel with a 180-degree rotation. **c** Overall interaction of the Pfam01383 (CpcD) domain of FNR (orange) with the Pfam00427 domain of CpcC2 (cyan). **d** The interface area of CpcC2 with the relevant residues shown. **e–g** Three representative areas of interactions between the CpcD domain of FNR and Pfam00427 of CpcC2. The distances are in angstrom.

the Pfam01383 domain of CpcC2 interacts with Pfam00427 domains of CpcC1. The binding of CpcC1 to CpcL is mediated by an extensive interface, involving the long N-terminal loop (G219-G233, $Loop_{C1}$), the loop between β1 and β2 (I253-K259, $Loop_{12}$), and the helix 9 (α9) of the Pfam01383 domain of CpcC1. These two loops fold towards CpcL and interact from two opposite sides. This tight interface is mediated by a large number of residues of CpcC1 and CpcL, including both polar and hydrophobic sidechains (Fig. 2a, b). Sequence alignment of CpcC1 and CpcC2 shows that while the overall sequences of the two proteins are very similar to each other, some residues of CpcC1 involved in the interactions with CpcL are not conserved in CpcC2 (Supplementary Fig. 4a). Superimposition of the Pfam01383 domains of CpcC1 and CpcC2 (Fig. 2b) shows that CpcC1 could have a more optimal interaction with CpcL than CpcC2 because of the distinct conformations of them in the two above-mentioned loop regions. In fact, sequence comparison shows that CpcC1 has a longer domain linker (16 residues more) than CpcC2, which is upstream

one of the CpcL-interacting loops (Supplementary Fig. 4a). This paralogue difference should also contribute to the preference of CpcL for CpcC1. In the CpcL-PBS structure, the interaction of the Pfam01383 domain of CpcC2 with the Pfam00427 domain of CpcC1 in CpcL-PBS is the same as that in the peripheral rods of CpcG-PBS[11–13]. Structurally, due to a shorter inter-domain linker, the two domains of CpcC2 are differently orientated compared to that in CpcC1: The Pfam01383 domain in CpcC2 is seen to have a 250° rotation (Supplementary Fig. 4b), leading to a slightly off-axial position of the middle hexamer of CpcL-PBS (Fig. 1a).

At the membrane-distal end of CpcL-PBS, the CpcD domain of $FNR_{3D}$ is unambiguously identified and it interacts with the Pfam00427 of CpcC2 mostly by polar interactions (Fig. 2c–g). This structural basis of the attachment of the CpcD domain of $FNR_{3D}$ to the Pfam00427 domain of a CpcC protein in a PC hexamer is important for our understanding of CpcL-PBS's role in NDH-PSI-PBS supercomplex formation and in cyclic electron flow[28–31]. The structures of the other two

domains of FNR$_{3D}$ in the CpcL-PBS could not be determined due to the long flexible inter-domain linker region[21,22]. We noticed that the CpcD protein located at distal ends of the peripheral rods of the CpcG-PBS from *Synechocystis* 6803[13] and the CpcD domain of FNR$_{3D}$ possess a similar binding pattern with the Pfam00427 domain of CpcC2 (Supplementary Fig. 5). Therefore, more study is needed to understand why no FNR$_{3D}$ is observed on the peripheral rods of the cryo-EM structures of CpcG-PBS.

### Chromophore conformation in CpcL-PBS

The overall distribution of bilins in CpcL-PBS is the same as that of a peripheral rod in CpcG-PBS[11,12] (Fig. 3a) and the bilins are in close contact with linker proteins (Fig. 3b). It has been suggested that protein-bilin interaction plays a crucial role in unidirectional energy transfer within the rods of CpcG-PBS[11], but the bilin conformations in the rods in intact PBS have not been carefully examined so far. Because the CpcL-PBS reported here and by Liu et al.[24] has a room temperature fluorescence emission peak at 668 nm, which is significantly red-shifted than that from any CpcG-related PC rods[26,27], it provides an excellent system for understanding the structural mechanism of energy transfer within PBS rods and to PSI. The high resolution of the cryo-EM CpcL-PBS structure allows us to compare the bilin conformation changes brought by protein-bilin interactions, which has been observed in the terminal emitter ApcD[32] and the phycobiliproteins from far-red light adapted species[33].

All bilins in CpcL-PBS are phycocyaninbilin (PCB), which is an open chain tetrapyrrole covalently linked to PC at ring A. The conformation of a bilin is defined by the planes of its four pyrrole rings and the three angles between rings. Seven different representative bilins, four from the CpcL-PBS and three from CpcG-PBS, are compiled for conformational comparison (Fig. 3c, d and Supplementary Fig. 3e–h). The majority of bilins in CpcL-PBS have a conformation like the bilin $^{2II}\beta^{82}_3$ (Fig. 3c, d and Supplementary Fig. 3h): the angle between rings A and B is between 26°–30°; the angle between rings B and C is smaller than 10°; the angle between rings C and D is larger than 35°. For the three $\beta^{82}$

bilins at the bottom trimer of CpcL-PBS, the conformations are different from that of $^{2II}\beta^{82}_3$: the angles between rings C and D are significantly reduced, which indicates that the planarity of these bilins is greatly enhanced. Because the ring A in these three bilins is near their β-PC and away from residues of CpcL, the angles between ring A and ring B remain almost unaltered. The most flattened bilin in CpcL-PBS is $^{II}\beta^{82}_2$. Its angles between rings A and B, B and C, and C and D are 26°, 7°, and 8°, respectively (Fig. 3c, d and Supplementary Fig. 3f). The terminal emitter ApcD has a fluorescence emission peak at 680 nm and its rings B, C and D are nearly coplanar with the three respective angles of 25°, 7°, and 2°, respectively[32] (Fig. 3c, d). Therefore, the bilin $^{II}\beta^{82}_2$ is expected to have a significant red-shift in absorption and fluorescence emission peaks. The bilins $^{II}\beta^{82}_1$ and $^{II}\beta^{82}_3$ are less but still significantly flattened and the angles between rings C and D are 21° in $^{II}\beta^{82}_1$ and 25° in $^{II}\beta^{82}_3$. Since the bilin $^{II}\beta^{82}_1$ has extensive interaction with CpcL (Fig. 3b), it is expected to have a significant red-shift in absorption and fluorescence emission. The flattened bilins reported here is not limited to CpcL-PBS. The bilin $^{II}\beta^{82}_{1G}$ ("G" for CpcG) from the peripheral rod of *Synechocystis* CpcG-PBS, which is equivalent to the bilin $^{II}\beta^{82}_1$ of CpcL-PBS, has the three respective angles of 30°, 11°, and 15°, very similar to that of the bilin $^{II}\beta^{82}_1$ of CpcL-PBS (Fig. 3d and Supplementary Fig. 3e). The bilin $^{II}\beta^{82}_{2G}$, which is equivalent to the bilin $^{II}\beta^{82}_2$ in position, has the three respective angles of 32°, 10°, and 20°, significantly less flattened than $^{II}\beta^{82}_2$ of CpcL-PBS (Fig. 3c, d). The bilin $^{II}\beta^{82}_{1G}$ of CpcG-PBS is likely the energy trap of the peripheral rods for energy transfer from rods to core[11] while $^{II}\beta^{82}_2$ of CpcL-PBS is a candidate as the trap for energy transfer from CpcL-PBS to PSI.

### Interactions of CpcL with chromophores

Each αβ monomer of CpcL-PBS has three covalently attached chromophores, one on α subunit (α84) and two on β subunits (β82 and β152), similar to the PC αβ monomers of the peripheral rods of CpcG-PBS[4]. Located in the center of the PC hexamer near the thylakoid membrane, CpcL is close to four bilins and extensively interacts with the three bottom β82 bilins ($^{II}\beta^{82}_1$, $^{II}\beta^{82}_2$, and $^{II}\beta^{82}_3$) (Fig. 3b). Spectroscopic studies

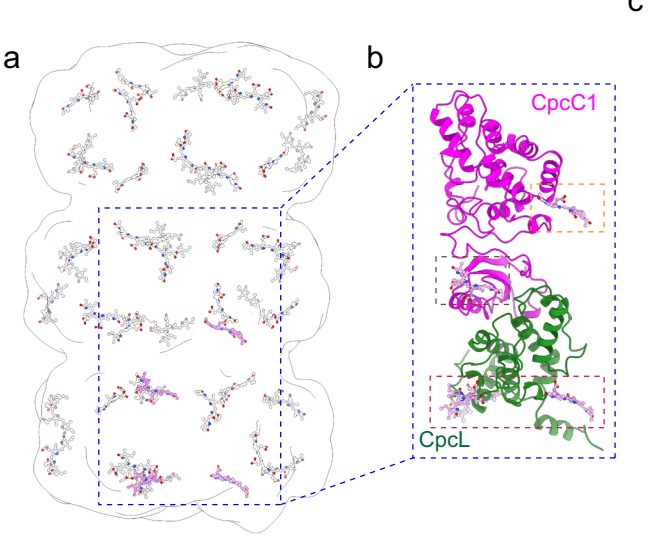

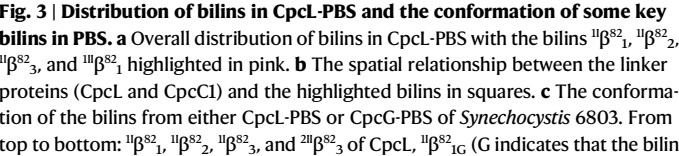

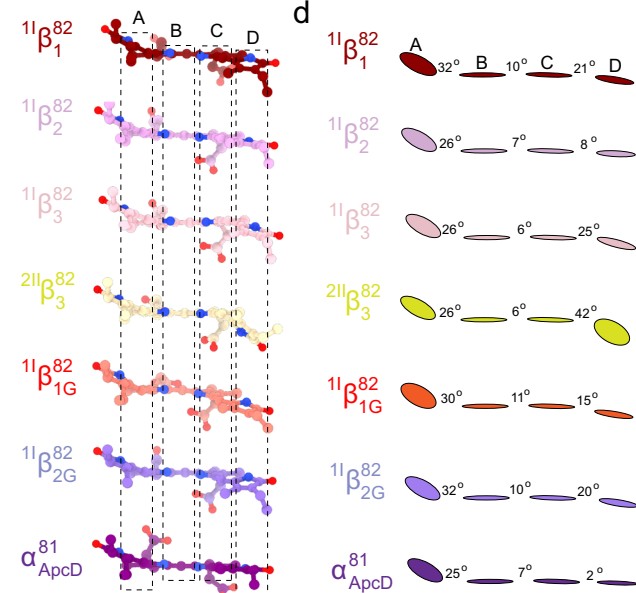

**Fig. 3 | Distribution of bilins in CpcL-PBS and the conformation of some key bilins in PBS. a** Overall distribution of bilins in CpcL-PBS with the bilins $^{II}\beta^{82}_1$, $^{II}\beta^{82}_2$, $^{II}\beta^{82}_3$, and $^{III}\beta^{82}_1$ highlighted in pink. **b** The spatial relationship between the linker proteins (CpcL and CpcC1) and the highlighted bilins in squares. **c** The conformation of the bilins from either CpcL-PBS or CpcG-PBS of *Synechocystis* 6803. From top to bottom: $^{II}\beta^{82}_1$, $^{II}\beta^{82}_2$, $^{II}\beta^{82}_3$, and $^{2II}\beta^{82}_3$ of CpcL, $^{II}\beta^{82}_{1G}$ (G indicates that the bilin

is from CpcG-PBS) and $^{II}\beta^{82}_{2G}$ of the peripheral rod1, and the bilin α84 of the terminal emitter ApcD of the CpcG-PBS core. The coordinates of bilins from CpcG-PBS of *Synechocystis* 6803 were from a recent cryo-EM study[13]. **d** Schematic presentation of bilins from rings A to D in panel (**c**). The angles between the adjacent pyrrole planes are shown.

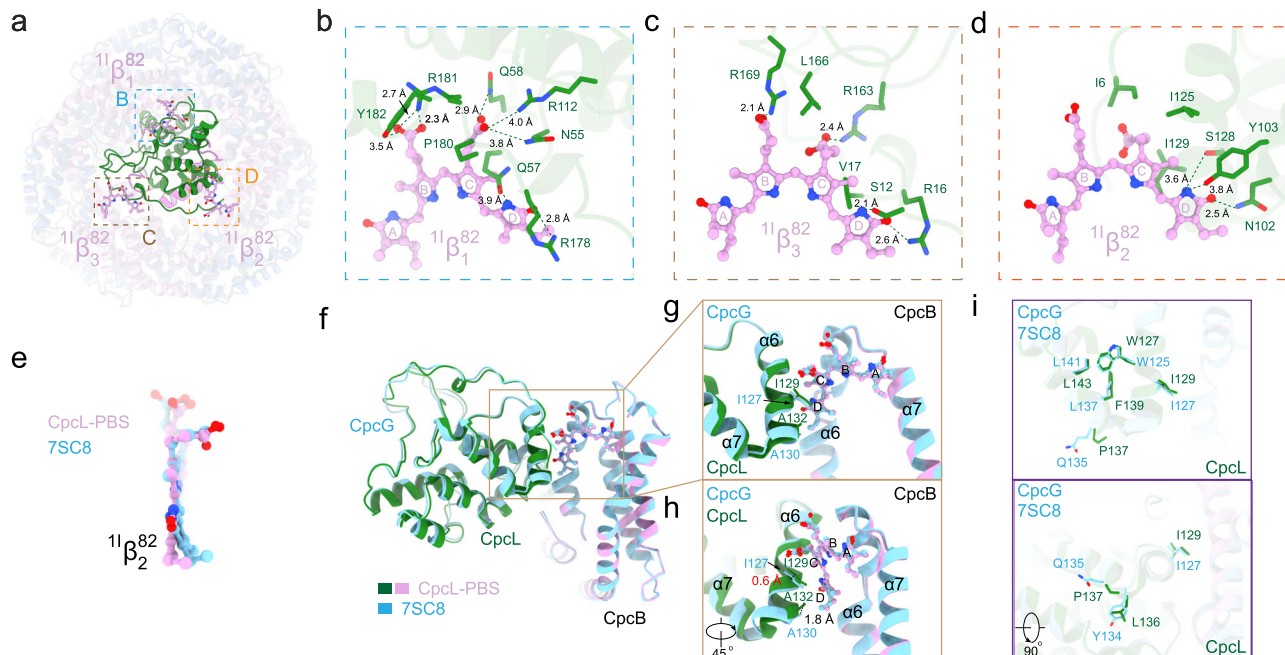

**Fig. 4 | Interactions of CpcL with the bilins β[82] at the bottom trimer of CpcL-PBS. a–d** Bottom view of CpcL-PBS, showing the CpcL (green) and the three β[82] bilins, $^{II}β^{82}_1$ (square **b**), $^{II}β^{82}_2$ (square **c**), and $^{II}β^{82}_2$ (square **d**), of CpcL-PBS. Interaction of key amino acid residues of CpcL (green) with $^{II}β^{82}_1$ (panel **b**) $^{II}β^{82}_2$ (panel **c**) and $^{II}β^{82}_3$ (panel **d**). The bilins (open chain tetrapyrrole) are in pink with the four pyrrole rings labeled A through D. Oxygen atoms are in red, nitrogen atoms are in blue. The dashed lines indicate interactions and the distances between the atoms are in Angstroms. **e** Structural comparison of bilin $^{II}β^{82}_2$ from CpcG-PBS rod (PDB 7SC8)[13] and CpcL-PBS. Alignment was done with the CpcB subunit as reference, and the RMSD of all non-hydrogen atoms on the ring D is 0.5 Å between two bilins. **f** Structural comparison of CpcG-CpcB (PDB 7SC8) and CpcL-CpcB complexes by aligning their subunit CpcB. The environment around bilin $^{II}β^{82}_2$ is highlighted in brown solid rectangle box. **g** Zoom-in view of the environment around bilin $^{II}β^{82}_2$. The difference between CpcL and CpcG lies in α6 and α7 helices. **h** A 45° rotated view of (**g**). The distance between the residues that regulate the conformation of their bilin $^{II}β^{82}_2$ are highlighted in red. **i** Residues that are responsible for modulating the conformation of this α6-α7 region.

have shown that the β[82] bilins, which are located interior of the PC trimer rings, function as acceptors of excitons while the other two bilins of a PC trimer act as donors[4]. When PC trimers form PBS rods with the help of linker proteins, especially the CpcG linker, the fluorescence emission is red-shifted[26,27], suggesting that the linkers could modulate the energy state of bilins. Cryo-EM structures of cyanobacterial CpcG-PBS show that the CpcG linker provides aromatic residues surrounding the bilin $^{II}β^{82}_1$, which transfers absorbed light energy from rods to PBS cores[11].

Comparison of the surrounding residues of the $^{II}β^{82}_1$ bilin from CpcG-PBS rod and CpcL-PBS shows that more interactions could be observed between the bilin $^{II}β^{82}_1$ and CpcL than that between the bilin $^{II}β^{82}_{1G}$ and CpcG (Fig. 4a, b and Supplementary Fig. 6a), and the interactions of the bilin $^{II}β^{82}_1$ with CpcG and CpcL are slightly different, especially in the vicinity of ring D (Supplementary Fig. 6a). Sequence alignment also reveals the divergence of the residues around ring D, such as Q57 and R178 of CpcL (Supplementary Fig. 6d). Three positively charged arginine residues (R178, R112, and R181) of CpcL are within 4 Å from the bilin $^{II}β^{82}_1$ (Fig. 4b and Supplementary Fig. 6a), and they either interact with the carboxylic acid groups on rings B and C or the oxygen on ring D. In addition, N55, Q57, Q58, and Y182 are capable of forming hydrogen bonds with either the nitrogen or oxygen atoms of $^{II}β^{82}_1$ (Fig. 4b and Supplementary Figs. 6a, 7a). The interaction between the bilin $^{II}β^{82}_3$ and CpcL is also extensive and involve three arginine residues (R16, R163, and R169) (Fig. 4c and Supplementary Figs. 6b, 7c). In contrast, the interactions between CpcL and $^{II}β^{82}_2$ do not involve any arginine residues (Fig. 4d and Supplementary Figs. 6c, 7b). The residues of CpcG that interact with $^{II}β^{82}_2$ and $^{II}β^{82}_3$ are highly conserved or invariant in CpcL (Supplementary Fig. 6b, c).

The CpcL-PBS has the fluorescence emission peaks at 648 nm and 668 nm at room temperature[24,25] (Supplementary Fig. 1g), the former is at the same position of isolated PC rods with linkers, while the latter is significantly red-shifted compared to any PC fluorescence emission. As mentioned above, the bilin $^{II}β^{82}_2$ but not $^{II}β^{82}_1$ may account for the redshift of fluorescence emission, since the bilin $^{II}β^{82}_2$ is flattened the most in CpcL-PBS. The environment of the bilin $^{II}β^{82}_2$ provided by β-PC is the same in CpcG-PBS rods and CpcL-PBS (Supplementary Fig. 7b), the difference in bilin conformation (Fig. 3c, d) between the bilins $^{II}β^{82}_2$ from CpcG-PBS and CpcL-PBS is probably generated by subtle difference of their surroundings created by CpcG or CpcL. Interestingly, key residues that are involved in the interaction of the bilins $^{II}β^{82}_2$ with CpcG and CpcL are identical (Fig. 4d and Supplementary Fig. 6c). Superimposition of the two bilins using the CpcB subunit as the reference of alignment reveals an RMSD of 0.5 Å for the ring D of the bilins and an apparent rotation for the ring D (by ~10°) (Fig. 4e). Compared to CpcG, no significant difference was observed in CpcL except for a region of two connected helices next to $^{II}β^{82}_2$ (α6 and α7) (Fig. 4f). In fact, the α6 of CpcG or CpcL and the α6 of CpcB clamp the bilin $^{II}β^{82}_2$ and should be able to modulate the conformation of ring D (Fig. 4g, h). In this region, an important difference is between I129 of CpcL and I127 of CpcG. Both residues are located next to the ring D of $^{II}β^{82}_2$ in their respective PBS structures (Supplementary Fig. 7b), but I129 of CpcL in CpcL-PBS has a -0.6 Å shift towards the ring D compared to I127 of CpcG in the CpcG-PBS (Fig. 4g, h). In this helical turn region (W125-L141 of CpcG and W127-L143 of CpcL), a few residues differ in identity between CpcL and CpcG, including L136, P137, and F139 of CpcL (Fig. 4i and Supplementary Fig. 6d). These residues are located in the turning region of the two helices, and the equivalent positions of them in CpcG are Y134, Q135, and L137, respectively (Fig. 4i). Apparently, these sequence differences (Supplementary Fig. 6d), especially a proline P137 in CpcL, generate local conformational differences to impact on the orientation of α6, which in turn

modulates the conformation of the bilin $^{II}\beta^{82}{}_2$. Therefore, it is our suggestion that these key residues in CpcL are responsible for providing spatial restriction on the bilin, leading to a rotation of the ring D and reduction of the angle between rings C and D.

The CpcL extends into the upper half of the bottom hexamer of CpcL-PBS and interacts with the bilins in the second trimer layer. For example, S70 and Q71 of CpcL could form hydrogen bonds with ring D while K73 could form hydrogen bond with ring C of $^{II}\beta^{82}{}_1$ (Supplementary Fig. 8). Besides CpcL, the linker proteins CpcC1, CpcC2 and the CpcD domain of FNR also participate interactions with bilins within CpcL-PBS (Supplementary Fig. 8) and these interactions could be important to bilin stabilization and energy transfer.

## Ultrafast spectroscopic study of energy transfer within CpcL-PBS

Ultrafast fluorescence spectroscopy has been used to study energy transfer within CpcL-PBS of *Synechocystis* 6803[25] and energy transfer from CpcL-PBS to PSI of *Anabaena* 7120[34], and both studies revealed the importance of the "red PCB" species of CpcL-PBS in energy transfer. To understand energy transfer dynamics, exciton migration among the bilins of CpcL-PBS and identity of the red PCB, femtosecond time-resolved transient absorption spectroscopy was performed with intact CpcL-PBS (Fig. 5a–c). In the transient absorption experimental setup, the excitation wavelength was set at 590 nm with a pulse duration of 100 fs. The as-acquired transient absorption spectra at different time delay are presented in Fig. 5a. Both ground state bleaching and stimulated emission contributed to the negative absorption while transient absorption due to excited state of bilins led to positive absorbance change. Four decay components with different time constants are obtained (Fig. 5b) by global analysis of the spectra. The fast component $P_1$ with a decay time of 3.6 ps has an absorption bleaching at 631 nm and a positive absorption peak at

671 nm; the second fast component $P_2$ has absorption bleaching at 637 nm and a positive absorption at 675 nm. The components $R_S$ and $R_L$ have an intermediate decay time of 200 ps, and their absorption bleaching are at 644 nm and 668 nm, respectively. The slow component T with a decay time of >1200 ps has an absorption bleaching at 669 nm. Global dynamics analysis shows that energy migration within CpcL-PBS occurs sequentially and in a picosecond time range (Fig. 5c):

$$P_1(3.6\,ps)/P_2(25\,ps) \rightarrow R_S/R_L(200\,ps) \rightarrow to\ T\ (>1200\,ps)$$

Here the component T is the red-PCB that has a photo-induced bleaching at 669 nm and a long decay time constant.

The energy transfer of CpcL-PBS is further studied by picosecond time-resolved fluorescence spectroscopy (Fig. 5d–f). The time-wavelength 2D maps of CpcL-PBS excited at 565 nm were obtained in different time ranges (Supplementary Fig. 9) and global analysis was performed on the acquired time-resolved fluorescence spectra. The decay-associated spectra (DAS) and the species-associated spectra (SAS) are presented in Fig. 5d and Fig. 5e, respectively. Three species-associated spectra with their respective decay time constant of 101 ps, 401 ps, and 1999 ps were obtained and the SAS with a time constant of 401 ps is composed of two components, one with a peak at 651 nm and the other at 669 nm. The dynamics of these components in energy transfer (Fig. 5f) confirms the following energy transfer scheme:

$$C_{645nm} \rightarrow C_{651nm}/C_{669nm} \rightarrow C_{672nm} \ (C: component)$$

With the determination of the cryo-EM structure of CpcL-PBS and of the components in energy transfer by ultrafast spectroscopy, the energy transfer process from a regular PCB to the "red PCB" within CpcL-PBS is deduced. In femtosecond time-resolved transient

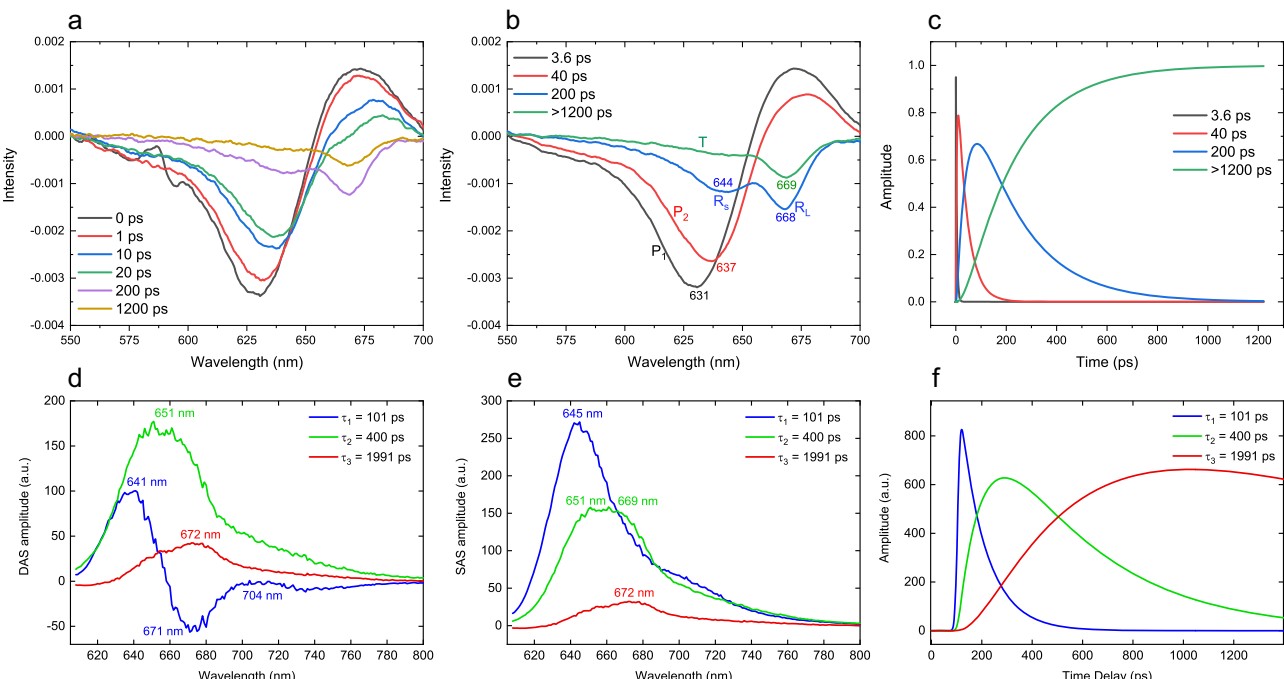

**Fig. 5 | Ultrafast absorption and fluorescence spectroscopic analysis of CpcL-PBS. a** Femtosecond time-resolved transient absorption spectra between 550 nm and 700 nm. The excitation wavelength is set at 590 nm and duration of the excitation laser pulse is 100 fs. Six time-resolved spectra of 100 fs, 1 ps, 10 ps, 20 ps, 200 ps, and >1200 ps are shown. **b** Four species-associated absorption spectra (SAS) are obtained from global analysis of the transient absorption spectra: $P_1$ (3.6 ps), $P_2$, (40 ps), $R_S$ (200 ps), $R_L$ (200 ps), and T (>1200 ps) and their bleaching is

at 631 nm, 637 nm, 644 nm, 668 nm, and 669 nm, respectively. **c** Kinetics of the four corresponding species-associated components. **d** Dynamics-associated difference fluorescence spectra (DAS) of CpcL-PBS excited at 565 nm; fluorescence is recorded with a streak camera. **e** Species-associated emission spectra (SAS) of CpcL-PBS. Peak wavelengths of each component are shown. **f** Kinetics of the three corresponding species-associated decay components.

absorption spectra, the fastest decay component $P_1$ has a bleaching peak at 631 nm and it is assigned to the bilins PCB $\alpha^{84}$ and $\beta^{155}$ as these bilins have little interaction with linker proteins and are expected to have the shortest absorption bleaching wavelengths. The component $P_2$ has a bleaching peak at 637 nm and it is assigned to all PCB $\beta^{82}$ except for the three bilins located at the bottom PC trimer. These bilins, located inside the cavities of the hexamers, interact with the linker proteins (Fig. 4 and Supplementary Fig. 7) and their absorption bleaching are expected to be red-shift[26,27]. The component $R_S$ is assigned to the bilin $^{II}\beta^{82}_3$ because it has an enhanced planarity (Fig. 3) and interacts with CpcL (Fig. 4 and Supplementary Fig. 6). The components $R_L$ and $R_S$ have the same decay constant of 200 ps, but $R_L$ has a significantly red-shifted absorption bleaching at 668 nm. The bilin $^{II}\beta^{82}_1$ has a similarly enhanced planarity as $^{II}\beta^{82}_{1G}$, but is engaged more interaction with the linker protein than either $^{II}\beta^{82}_{1G}$ or $^{II}\beta^{82}_3$ (Fig. 4 and Supplementary Fig. 6). Therefore, $R_L$ is tentatively assigned to $^{II}\beta^{82}_1$. Because the bilin $^{II}\beta^{82}_2$ has the most enhanced planarity and extensive interaction with CpcL, the component T is assigned to $^{II}\beta^{82}_2$, the red PCB of CpcL-PBS. In picosecond fluorescence spectra, the fluorescence emission of all PCB except for the three bilins $^{II}\beta^{82}$ merged as the spectrum of $C_{645nm}$ (100 ps); the bilins $^{II}\beta^{82}_3(R_S)$, $^{II}\beta^{82}_1$, $(R_L)$ give rise to the spectra of $C_{651nm}/C_{669nm}$ (401 ps); and the bilin $^{II}\beta^{82}_2$ (T) give rise to the spectrum of $C_{672nm}$ (1999 ps). The structure and the time-resolved fluorescence emission spectrum of the bilin $^{II}\beta^{82}_2$ (component T) has special characters of red-PCB like the bilin $\alpha^{84}$ in ApcD[35], clearly demonstrating here that CpcL-PBS has structural features for the formation of a terminal emitter-like red-PCB for direct energy transfer to PSI. Our study also shed light on energy transfer mechanisms in other types of PBS found in red algae and cyanobacteria, including *Acaryochloris* sp.[36].

## Methods

### Strains and culture conditions

The wild-type strain of *Synechocystis* sp. PCC 6803 (*Synechocystis* 6803) and its mutant strain are grown in BG11 medium[37] at 25 °C with light intensity of 50 μmol photons $m^{-2} s^{-1}$. The medium for the growth of the mutant strain was supplemented with 10 μg $ml^{-1}$ erythromycin and 10 mM glucose. The cultures were bubbled gently with air plus 1% $CO_2$.

### Construction of ΔapcAB mutant strain

The ΔapcAB mutant strain was constructed as described in Ajlani et al.[38] with some modifications. The upstream and downstream region of *apcAB* gene and a DNA fragment encoding an erythromycin-resistant gene were amplified by polymerase chain reaction (PCR) using primer pairs of P1/P2, P3/P4, and P5/P6, respectively (Supplementary Table 1). The above sequences were ligated by fusion PCR. The resultant PCR products were transformed into the wild-type *Synechocystis* 6803 strain. Segregation of the ΔapcAB mutant was verified by PCR.

### Isolation of CpcL-PBS

CpcL-PBS was isolated from the ΔapcAB mutant strain according to Liu et al.[24], or with some modifications as follows. All operations were performed at 22 °C. Briefly, the cells were broken in 0.75 M K/Na-PO₄ buffer at pH 7.0 with 1 mM phenylmethylsulfonyl fluoride by passing through three times of a French press at 7000 psi. After 0.5 h of incubation with 0.5% n-dodecyl-β-d-maltoside (DDM) (w/v), samples were centrifuged at 20,000 × g. The supernatant was loaded immediately onto a sucrose gradient. The sucrose gradients were made from buffer A (0.75 M K/Na-PO₄ buffer, 0.0125% glutaraldehyde (w/v), 0.03% β-DDM (w/v), pH 7.0) by adding sucrose to these concentrations: 0.4, 0.55, 0.7, 0.85, 1.0, and 1.5 M. The samples were centrifuged at

370,000 × g for 16 h using a TLA110 rotor on Beckman Optima MAX-XP centrifuge. CpcL-PBS samples were collected at the interface of 0.85 M to 1.0 M sucrose. The sucrose of purified CpcL-PBS samples was removed by ultrafiltration with 30 kD Millipore centrifugal filters. Absorption spectra and fluorescence emission spectra were obtained as described previously[11].

### Cryo-EM sample preparation and data collection

Before preparing grids for cryo-EM, CpcL-PBS samples were concentrated to -0.75 mg $ml^{-1}$. Aliquots (3.5 μl) of samples were loaded onto glow-discharged (20 s) holey carbon films copper grids (Quantifoil R1.2/1.3, +2 nm C membrane, Cu 300 mesh) and waited for 60 s. 3.5 μl aliquot of a buffer (50 mM Tris pH 7.0, 0.01% β-DDM (w/v)) was then added twice to the grid and mixed immediately with the sample before vitrification (to dilute the high concentration of phosphate salts). After blotting for 2 s, the grid was plunged into liquid ethane with an FEI Vitrobot Mark IV (18 °C and 100% humidity). Cryo-grids were first screened in a Talos Arctica operated at 200 kV (equipped with an FEI CETA camera). Grids of good quality were transferred to a FEI Titan Krios operated at 300 kV with Gatan BioQuantum GIF/K3 direct electron detector for data collection.

Movies were recorded with a BioQuantum GIF/K3 direct electron detector (Gatan) in the super-resolution mode at a nominal magnification of ×81,000, with an exposure rate of 17.9 $e^-/Å^2$ per second using the EPU 2 software. A GIF Quantum energy filter (Gatan), with a slit width of 20 eV was used at the end of the detector. The defocus range was set from −1.0 to −1.8 μm. The total exposure time was 3.84 s and 32 frames per image were acquired with a total electron exposure of ~60 $e^-/Å^2$. Statistics for data collection are summarized in Supplementary Table 2.

### Image processing

To determine the 3D structure of the CpcL-PBS complex, Two batches of movie stacks were recorded. Raw movie frames were aligned and averaged into motion-corrected summed images with a pixel size of 1.07 Å by MotionCor2[39]. The Gctf program (v1.06)[40] was used to estimate the contrast transfer function (CTF) parameters of each motion-corrected image. All the following data processing was performed with Relion-3.1[41]. 1350 particles were manually picked and subjected to 2D classification to generate templates for automatic particle picking. Then autopicking was done to the images that were manually selected for treatment. The picked particles were subjected to one round of 2D classification, and high-quality particles were further selected for subsequent 3D classifications. Initial model was generated from these particles. After multiple rounds of 3D classification, relatively homogeneous particles were selected for 3D refinement, resulting in a map with a 2.64 Å overall resolution after mask-based post-processing, based on the gold-standard FSC 0.143 criteria. The local resolution map was analyzed using ResMap[42] and displayed using UCSF Chimera[43]. Workflow of data processing was illustrated in the Supplementary Fig. 2.

### Model building and refinement

Crystal structure of a PC αβ monomer from *Synechocystis* 6803 (PDB code 4F0T)[44] was used as the initial template for PC hexamer modeling. The atomic models of CpcL, CpcC1, CpcC2, and PetH were adopted from AlphaFold databases[45]. All structures were firstly docked into the density map using UCSF chimera[43] and then manually adjusted in COOT[46]. The final atomic models were refined in real space using PHENIX[47] with secondary structure and geometry restraints applied. The final atomic models were evaluated using Molprobity[48] and the statistics of data collection and model validation were included in Supplementary Table 2.

## Femtosecond time-resolved transient absorption spectroscopy

The details of the home-built ultrafast absorption spectroscopy setup has been described elsewhere[49]. Briefly, a Ti:sapphire laser (Hurricane, Spectra-Physics, USA) delivered 100 fs (FWHM) pulses at a central wavelength of 800 nm at a repetition rate of 1 kHz. Then the pulses were split into two beams using a beam splitter: one was used to generate a supercontinuum white light which was used as the probe light. The other was used to pump a home-built non-collinear optical parametric amplifier (NOPA) for generating excitation source of tunable wavelength in the visible region. In current experiment, the excitation wavelength was set at 590 nm. The excitation energy was ~10 nJ per pulse with a spot size of ~120 μm. The relative polarization of the pump and probe pulses was set at the magic angle. The sample was placed into a 1-mm-thick fused silica flow cell in order to avoid sample damage, the optical density (OD) of the sample was set at 0.26 at its maximum absorption wavelength at 620 nm. The decay kinetics was scanned at a temporal resolution of 0.02 ps per step at early stage while a varied and larger time-delay steps were used at later stage. The time-resolved difference absorption spectral data were averaged over 30 times to obtain a better signal-to-noise ratio.

## Femtosecond time-resolved fluorescence spectroscopy

For the time-resolved fluorescence measurements, the sample here was placed into a 1-mm-thick fused silica cuvette with an optical density of 0.24 at 620 nm. Femtosecond amplifier (Spectra Physics, USA) delivered 100 fs pulses at a repetition rate of 5 kHz. The excitation wavelength was set at 565 nm from a commercialized tunable optical parametric amplification system (TOPAS, Spectra Physics) pumped by the femtosecond amplifier. Pulse energy of 5 nJ per pulse with a spot size of ~200 μm was used to excite the sample. A high-quality 600 nm long-pass band filter was used to block the residual excitation pulse. The emitted fluorescence light was collected and focused into a spectrograph and measured by a streak camera 5200 (XIOPM, China) operating at 1.4 ns or 5 ns scanning ranges with a time resolution of ~30 ps or ~200 ps, respectively. The spectral window recorded by the CCD of streak camera was 260 nm in the measurements, covering the fluorescence wavelength of the sample. The time-resolved fluorescence data were averaged over 10 single measurements.

Global fitting was used to resolve the individual spectral components from the congested spectrum. The program was implemented based on LabView. We employed the method of singular value decomposition (SVD) according to the previous report[49]. DAS (dynamics-associated different spectra) is a kinetic-dependent differential spectrum that can be obtained from global fitting directly. The species-associated emission spectra (SAS) and the corresponding kinetics could be resolved by establishing a model based on the result from DAS as SAS is determined by the analytical model we have chosen. Here we firstly obtained the lifetime of each component from DAS, and then we attribute each process to the corresponding pigments involved in energy transfer model based on the pigment molecules resolved in the cryo-EM structure:

$$P_1(3.6\,ps)/P_2(25\,ps) \rightarrow R_S/R_L\,(200\,ps) \rightarrow to\ T\,(>1200\,ps)$$

More details about global fitting could be found in our previous publications[50].

## Reporting summary

Further information on research design is available in the Nature Portfolio Reporting Summary linked to this article.

## Data availability

The data that support this study are available from the corresponding authors upon request. Atomic coordinate of the CpcL-PBS structure has been deposited to the Protein Data (PDB) with the following accession code 8HFQ (CpcL-PBS). The corresponding cryo-EM map has been deposited to the Electron Microscopy Data Bank (EMDB) under the following accession number EMD-34724 (CpcL-PBS). The previously published structures of the rod of CpcG-PBS from Synechocystis 6803 can be accessed via accession code 7SC8. Supplementry Fig. 1e, f are provided as Source data file. Source data are provided with this paper.

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

## Acknowledgements

We thank the Core Facilities of Peking University School of Life Sciences for assistance with negative-staining electron microscopy; the National Centre for Protein Sciences at Peking University for assistance with sample prepration; the Cryo-EM Platform and the Electron Microscopy Laboratory of Peking University for cryo-EM data collection; the High-performance Computing Platform of Peking University for help with computation. This work was supported by the National Science Foundation of China (22027802 to Y.W., 32070203 and 91851118 to J.Z., 32270253 to Z.G.Z.), the Ministry of Science and Technology of China (2017YFA0503703 to J.Z.), and the Qidong-SLS Innovation Fund to N.G.

## Author contributions

N.G., Y.W., and J.Z. conceived the project and designed experiment; Z.D.Z., Z.G.Z., and C.D. constructed mutant; Z.D.Z., H.W., and Z.G.Z. isolated PBS and performed biochemical analysis; L.Z., Z.D.Z., and Z.G.Z. performed cryo-EM experiment; L.Z. and G.W. performed model building for structure determination; J.W., H.L., and H.C. performed ultrafast spectroscopic analysis of PBS; L.Z., Z.G.Z., N.G., G.W., Y.W., and J.Z. analyzed and interpreted data. L.Z., N.G., Y.W., and J.Z. wrote the manuscript with help from all authors.

## Competing interests

The authors declare no competing interests.
