## [Peer Review File · Nature Communications]

Cryo-EM and femtosecond spectroscopic studies provide mechanistic insight into the energy transfer in CpcL-phycobilisomesReviewers' Comments:

Reviewer #1:

Remarks to the Author:

The manuscript entitled "Cryo-EM and femtosecond spectroscopic studies provide mechanistic insight into the energy transfer in CpcL-phycobilisomes" by Zheng et al presents a structural and spectroscopic study of *Synechocystis* sp. PCC 6803 CpcL phycobilisomes. CpcL phycobilisomes are simpler versions of their CpcG counterparts, consisting only of the equivalent of CpcG rods. In contrast, CpcG phycobilisomes are massive assemblies comprised of both core and rod subunits. Still, both CpcL and CpcG phycobilisomes function as light harvesting complexes for photosystems and are therefore crucial to study in order to understand the mechanics of photosynthesis in cyanobacteria and other phycobilisome-containing organisms.

Concerning the cryo-EM part, resolving CpcL structure appears to me to be less of a challenge as solving CpcG phycobilisomes, but still an important task. Technically, the cryo-EM workflow is proper, but I am a bit concerned about the interpretation and conclusions that the authors are drawing from these structures (see in major comments).

I won't comment on the spectroscopy as I am not qualified to judge that part.

Importantly, I will not go through all the typos and mistakes, but I strongly suggest having a native speaker proofread the manuscript. There are several instances where sentences are not properly structured or tenses and conjugations are incorrect, which could lead to misunderstandings of the authors' intended meaning.

Major comments:

- Throughout the manuscript delete all instances of "atomic resolution" (e.g. in the abstract line 34), or "near atomic resolution". Individual atoms can only be resolved at resolutions of 1.2Å or below, which is beyond the resolution of the data presented in this study.
 - Figures need a bit more work to put everything in their proper context. In Figure 1 for instance, the orientation of CpcL relative to the membrane/photosystem should be indicated. Additionally, I feel like a structural comparison with CpcG rods as a main figure would really benefit the manuscript.
 - Line 105-107: "The transmembrane region of CpcL, which was predicted to anchor CpcL-PBS to the thylakoid membranes, was not resolved probably due to its unrestricted movement in the purified complexes." Is this what we see as a faint density in Extended Figure 2, on the left side of CpcL above panel e?
 - Line 138-139: "The structures of the other two domains of FNR3D in the CpcL-PBS could not be determined because they are highly flexible". What does AlphaFold predicts?
 - In Figure 3 the authors claim important conformational differences between CpcL bilins. The bilin models should be shown in their respective cryo-EM densities, in the main figure. I think Extended Data Figure 6 shows the bilins' densities, but the figure is of poor quality and almost no density is clearly visible. At 2.6Å, aromatic protein side chains (Tyr, Phe, Trp) should be clearly resolved, but I have a hard time seeing them here.
- Line 154, "atomic resolution of the cryo-EM CpcL-PBS structure allows us to compare the liblin conformation changes" – IF you had atomic resolution for sure, but here you do not have that kind of resolution.
- It is important because, when you make claims such as in line 227-229 "Superimposition of the two bilins reveals a 0.6-Å movement of ring D and a 12° angle change between ring C and D" – at claimed 2.6Å resolution, how confident are you about a 0.6Å shift?

Reviewer #2:

Remarks to the Author:

Zheng et al., reported the first structure of CpcL-PBS isolated from a mutant cyanobacterium in which CpcG-PBS is absent. The reddest bilin pigment and its chemical environments indicate the red shift

mechanism in this special type of PBS. Time-resolved absorption and fluorescence spectroscopy were also used to characterize the excitation energy transfer within CpcL-PBS. The experimental designs are reasonable, the data analysis and interpretation, however, need improvement as suggested as follows,

1. The stoichiometry of CpcC1, CpcC2, CpcCL. Zheng et al., observed that CpcC2 showed a much stronger band than that of CpcC1 in the Extended Data Fig. 1e, which seems consistent with a report from Liu et al., Ref 24 in which an elongated CpcL-PBS model was proposed. Please discuss it.

2. In Extended Data Fig. 1f, please label those particles potentially containing 3 hexamers that have been used for data refinement. My concern here is that CpcL-PBS could be very heterogenous in nature supported by Extended Data Fig. 1e and Liu et al. reports Ref 24, i.e. different CpcC2 copy numbers per CpcL-PBS. Focusing on those particles with potential 3 hexamers subsequently underestimated the complexity of CpcL-PBS structure.

3. In this manuscript, it's not clearly stated if CpcD subunit is present in the structure. It claimed that CpcD-like domain of FNR3D was observed in the distal end of CpcL. The questions here is that are all the CpcL-PBSs capped by CpcD domain of FNR3D or some of the CpcL-PBSs are capped by canonical PBS subunit CpcD. Which key amino acids (electron density) were used to define the identity of the CpcD domain of FNR3D rather than CpcD subunit. Please clarify. This question makes particularly important if "the other two domains of FNR3D in the CpcL-PBS could not be determined because they are highly flexible", page 6, second graph, line 6-7.

4. What are the electron density of 2.64 Angstrom model after 3D refinement in Extended Data Fig. 2.

We sincerely thank the two reviewers for their thoughtful suggestions and criticisms, which are very helpful for us to prepare the revision.

Reviewer #1 (Remarks to the Author):

The manuscript entitled “Cryo-EM and femtosecond spectroscopic studies provide mechanistic insight into the energy transfer in CpcL-phycoobilisomes” by Zheng et al presents a structural and spectroscopic study of *Synechocystis* sp. PCC 6803 CpcL phycoobilisomes. CpcL phycoobilisomes are simpler versions of their CpcG counterparts, consisting only of the equivalent of CpcG rods. In contrast, CpcG phycoobilisomes are massive assemblies comprised of both core and rod subunits. Still, both CpcL and CpcG phycoobilisomes function as light harvesting complexes for photosystems and are therefore crucial to study in order to understand the mechanics of photosynthesis in cyanobacteria and other phycoobilisome-containing organisms.

Concerning the cryo-EM part, resolving CpcL structure appears to me to be less of a challenge as solving CpcG phycoobilisomes, but still an important task. Technically, the cryo-EM workflow is proper, but I am a bit concerned about the interpretation and conclusions that the authors are drawing from these structures (see in major comments).

I won't comment on the spectroscopy as I am not qualified to judge that part.

Importantly, I will not go through all the typos and mistakes, but I strongly suggest having a native speaker proofread the manuscript. There are several instances where sentences are not properly structured or tenses and conjugations are incorrect, which could lead to misunderstandings of the authors' intended meaning.

Major comments:

- Throughout the manuscript delete all instances of “atomic resolution” (e.g. in the abstract line 34), or “near atomic resolution”. Individual atoms can only be resolved at resolutions of 1.2Å or below, which is beyond the resolution of the data presented in this study.

Suggestion is well taken. We have deleted them in the revision.

- Figures need a bit more work to put everything in their proper context. In Figure 1 for instance, the orientation of CpcL relative to the membrane/photosystem should be indicated.

Suggestion is well taken. We have updated the Fig. 1f (now Fig. 1g in the revision), in which the photosystem I and thylakoid membrane are shown.

Additionally, I feel like a structural comparison with CpcG rods as a main figure would really benefit the manuscript.

We thank the reviewer for this suggestion. We have added the figure of structural comparison between CpcL-PBS and CpcG rods as Fig. 1f and also included a short description in the text.

- Line 105-107: “The transmembrane region of CpcL, which was predicted to anchor CpcL-PBS to the thylakoid membranes, was not resolved probably due to its unrestricted movement in the purified complexes.” Is this what we see as a faint density in Extended Figure 2, on the left side of CpcL above panel e?

The reviewer is correct on this. On the CpcL structure predicted by AlphaFold, there is a single transmembrane helix at the C-terminus (Response Figure 1). There are two density blobs on the two terminal ends of CpcL-PBS (Above panel e of Extended Data Fig. 2). The purification buffer we used contained 0.5% DDM. The detergent micelle of CpcL-TM helix is on the left, whereas the density blob of the NADPH binding and FAD binding domains of FNR is on the right.

Response Figure 1 | Structure of CpcL predicted by AlphaFold. The N terminal Pfam00427 domain and C-terminal transmembrane helix are highlighted by magenta and orange dotted lines, respectively.

- Line 138-139: “The structures of the other two domains of FNR3D in the CpcL-PBS could not be determined because they are highly flexible”. What does AlphaFold predicts?

We have revised our inaccurate description about the cyanobacterial FNR. Most cyanobacterial FNR contain three successive domains: CpcD domain which is located in the cavity of CpcA-CpcB hexamer, FAD domain, an antiparallel beta barrel region that could bind the FAD cofactor, and the NADPH binding domain that contains an alpha helix-beta strand fold, where the NADP⁺ binds. While the FAD and NADPH binding domains are packed tightly, the CpcD domain is connected to them though a long flexible loop (Response Figure 2). This loop makes these two domains relatively flexible.

Therefore, in our structure, only the CpcD domain can be well defined and the densities of FAD and NADPH binding domains are highly fragmented. We have revised the text to make it more accurate in the revision.

Response Figure 2 | Structure of FNR predicted by AlphaFold.

- In Figure 3 the authors claim important conformational differences between CpcL bilins. The bilin models should be shown in their respective cryo-EM densities, in the main figure.

We thank the reviewer for this thoughtful suggestion. We have added the densities of CpcL bilins in Extended Data Fig. 3e-h.

I think Extended Data Figure 6 shows the bilins' densities, but the figure is of poor quality and almost no density is clearly visible. At 2.6Å, aromatic protein side chains (Tyr, Phe, Trp) should be clearly resolved, but I have a hard time seeing them here.

We have changed the density representation of the figures showing the local environment of the ring D of $^1\beta^{82}_1$, $^1\beta^{82}_2$ and $^1\beta^{82}_3$ (Response Figure 3). In the revised panels, these residues can be clearly seen.

Response Figure 3 | Local environment of ring D of the bilin $^1\beta^{82}_1$, $^1\beta^{82}_2$ and $^1\beta^{82}_3$ from *Synechocystis* 6803. a-c, Local environment of the ring D of the bilins $^1\beta^{82}_1$ (a),

$^1\beta^{82}_2$ (b) and $^1\beta^{82}_3$ (c). Residues from CpcL is colored as forest green and the CpcB is colored as pink, respectively. Bilins are color-coded same as in Fig. 3c.

Line 154, “atomic resolution of the cryo-EM CpcL-PBS structure allows us to compare the bilin conformation changes” – IF you had atomic resolution for sure, but here you do not have that kind of resolution.

It is important because, when you make claims such as in line 227-229 “Superimposition of the two bilins reveals a 0.6-Å movement of ring D and a 12° angle change between ring C and D” – at claimed 2.6Å resolution, how confident are you about a 0.6Å shift?

We fully understand the concern of the reviewer. As shown in Extended Data Fig. 3e-h in the revision, the bilins agree with their densities very well. At 2.6 Å resolution, most of the protein side chains are well resolved. In terms of model precision and accuracy, they should exceed the nominal resolution of 2.6 Å. We have used real-space refinement in the Phenix package to refine the model (with the restraints of ligands generated by eLBOW) and performed pairwise comparison using Chimera.

As shown in Fig. 4e, the bilins $^1\beta^{82}_2$ from CpcG-PBS rod (PDB 7SC8) and CpcL-PBS do exhibit different conformations. When the CpcB subunit was used as reference of alignment, the RMSD of all non-hydrogen atoms on the ring D is 0.52 between two bilins. The previous measurement of 0.6 Å was based on the distance of the hydroxyl oxygen on the ring D. We agree with the reviewer that we do not know the error in this measurement. In the revision, we have rephased the sentence to “Superimposition of the two bilins using the CpcB subunit as the reference of alignment reveals an RMSD of 0.5 Å for the ring D of the bilins and an apparent rotation for the ring D (by ~10°).”

Reviewer #2 (Remarks to the Author):

Zheng et al., reported the first structure of CpcL-PBS isolated from a mutant cyanobacterium in which CpcG-PBS is absent. The reddest bilin pigment and its chemical environments indicate the red shift mechanism in this special type of PBS. Time-resolved absorption and fluorescence spectroscopy were also used to characterize the excitation energy transfer within CpcL-PBS. The experimental designs are reasonable, the data analysis and interpretation, however, need improvement as suggested as follows,

1. The stoichiometry of CpcC1, CpcC2, CpcCL. Zheng et al., observed that CpcC2 showed a much stronger band than that of CpcC1 in the Extended Data Fig. 1e, which seems consistent with a report from Liu et al., Ref 24 in which an elongated CpcL-PBS model was proposed. Please discuss it.

We fully agree with the reviewer that the CpcL-PBS is heterogenous in nature. And we also observed more than three layers of hexamers in our negatively stained CpcL-PBS images, in particular, four layers (Response Figure 4). The structure of CpcL-PBS we determined should reflect the major form of CpcL-PBS in our purified samples.

We have added the discussion of CpcL-PBS heterogeneity in the revision and revised Extended Data Fig. 1f to highlight both three-layered and four or more layered particles. With that said, as shown in our 2D classification results, the three-layered form is clearly the predominant species of CpcL-PBS particles.

Response Figure 4 | Negative staining images of CpcL-PBS particles. The CpcL-PBS contained more than three layers hexamers are circled with blue solid line.

2. In Extended Data Fig. 1f, please label those particles potentially containing 3 hexamers that have been used for data refinement. My concern here is that CpcL-PBS

could be very heterogeneous in nature supported by Extended Data Fig. 1e and Liu et al. reports Ref 24, i.e. different CpcC2 copy numbers per CpcL-PBS. Focusing on those particles with potential 3 hexamers subsequently underestimated the complexity of CpcL-PBS structure.

We thank the reviewer for this suggestion. Based on our analysis, the number of particles with more than three layers of hexamers is very small. We could not produce high-resolution structures from them. We have followed the suggestion to add relevant discussion on the compositional heterogeneity in the revision.

3. In this manuscript, it's not clearly stated if CpcD subunit is present in the structure. It claimed that CpcD-like domain of FNR3D was observed in the distal end of CpcL. The questions here is that are all the CpcL-PBSs capped by CpcD domain of FNR3D or some of the CpcL-PBSs are capped by canonical PBS subunit CpcD. Which key amino acids (electron density) were used to define the identity of the CpcD domain of FNR3D rather than CpcD subunit. Please clarify. This question makes particularly important if "the other two domains of FNR3D in the CpcL-PBS could not be determined because they are highly flexible", page 6, second graph, line 6-7.

From the sequence alignment of CpcD and the CpcD-like domain of FNR, three regions could be used as landmarks to define the identity of the density (Response Figure 5a). First, the residue of S10/FNR is quite different from Y10/CpcD, because Y10 has a large side chain. And the Q13 of FNR is absent in CpcD (Response Figure 5b). Second, G36 and L37 of FNR are also very different from N35 and S36 of CpcD (Response Figure 5c). Third, T50, L53 and K54 of FNR are different from K49, Y52 and A53 of CpcD, respectively (Response Figure 5d). Our map clearly shows that the terminal protein should be FNR but not CpcD.

Response Figure 5 | Sequence and structure analysis of FNR subunits. a,

Sequence alignment of CpcD and CpcD-like domain FNR. These regions that determined

this density belongs to FNR are highlighted with magenta solid lines. **b-d**, Local densities of three representative regions are shown.

4. What are the electron density of 2.64 Angstrom model after 3D refinement in Extended Data Fig. 2.

We have prepared another figure (Extended Data Fig. 3) to show the densities of some representative subunits.

Reviewers' Comments:

Reviewer #1:

Remarks to the Author:

The revised manuscript by Zheng et al. was improved in respect to the first version and addresses all my original concerns and questions.

Reviewer #2:

Remarks to the Author:

The authors have addressed all my questions/suggestions. Thanks,